# Thinking Slow, Fast:
# Scaling Inference Compute with Distilled Reasoners

## Abstract

Recent advancements have demonstrated that the performance of large language models (LLMs) can be significantly enhanced by scaling computational resources at test time. A common strategy involves generating multiple Chain-of-Thought (CoT) trajectories and aggregating their outputs through various selection mechanisms. This raises a fundamental question: can models with lower complexity leverage their superior generation throughput to outperform similarly sized Transformers for a fixed computational budget? To address this question and overcome the lack of strong subquadratic reasoners, we distill pure and hybrid Mamba models from pretrained Transformers. Trained on only 8 billion tokens, our distilled models exhibit strong performance and scaling on mathematical reasoning datasets while being much faster at inference for large batches and long sequences. Despite the zero-shot performance hit due to distillation, both pure and hybrid Mamba models can scale their coverage and accuracy performance past their Transformer teacher models under fixed time budgets, opening a new direction for scaling inference compute.

## 1 Introduction

Reasoning in large language models (LLMs) has seen a significant boost in performance recently, largely driven by scaling inference compute. A key technique to enhance "reasoning" performance is the use of intermediate reasoning steps before producing a final answer, known as Chain-of-Thought (CoT) (Wei et al., 2023). Building on this, many test-time compute techniques often involve generating multiple CoTs (Wu et al., 2024; Snell et al., 2024) and selecting the best one. Although trained reward models can assign scores to the final output and even to the intermediate steps of the CoTs (Luo et al., 2024), leading to sophisticated selection schemes, even simple strategies, such as majority voting, can be surprisingly effective (Brown et al., 2024; Beeching et al., 2024).

However, these test-time compute techniques introduce significant challenges for LLM systems. Generating long CoT sequences or large batches of completions places substantial demands on memory and compute resources. Transformers, in particular, struggle with such workloads due to their linear memory scaling and memory-bound nature during generation. This raises an important question: *how should we optimize model architectures to best scale test-time compute?* In particular, can alternative architectures with faster and more efficient generation outperform current LLMs under fixed compute budgets? Addressing this problem could unlock new avenues for deploying reasoning models with different architectures, enabling them to run and scale more efficiently on hardware and environments with limited memory and compute.

Recent subquadratic architectures have training or prefill time linear in sequence length, and constant memory requirement (instead of linear) during inference. This enables up to $5\times$ higher inference throughput (Gu & Dao, 2024; Peng et al., 2023), as inference time for large batch size or long sequences is dominated by the loading of model states (KV cache or RNN states). Despite their efficiency, subquadratic models have not been extensively explored in reasoning tasks, primarily due to the lack of large-scale pretrained models compared to Transformer-based counterparts. Thus, it remains unclear whether: (1) scaling inference compute for subquadratic models improves reasoning performance, and (2) subquadratic models can match or exceed Transformers models under fixed compute budgets.

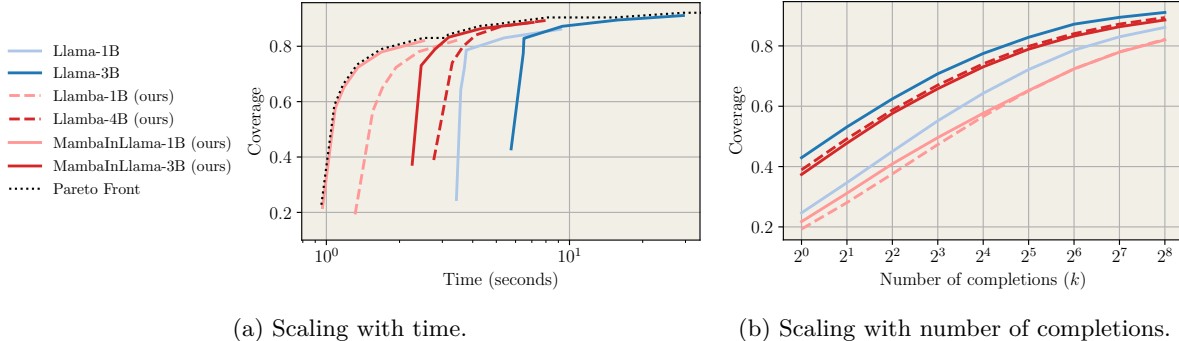

(a) Scaling with time.

(b) Scaling with number of completions.

Figure 1: **Distilled models have better coverage on MATH for most time budgets.** In **(b)**, we show the coverage as we increase the number of sampled answers $k$. Compared to their associated Llama baselines, our distilled models provide a lower coverage for a given $k$. In **(a)**, we now show the shortest time required to reach a given coverage. For each curve in **(b)**, we map the $k$-values on the x-axis to the time required to generate that many samples for each model. Ideally, we would want to reach the highest coverage for short time budget. For a given time budget, our distilled models can generate many more completions than their respective baselines. As such, the higher throughput of our models, shown in Figure 2, allows them to overcome their lower per-sample coverage. As a result, our models push the Pareto front forward for most time budgets.

In this work, we explore the reasoning capabilities of subquadratic architectures by distilling knowledge from pretrained Transformers into hybrid and pure Mamba models. To address the scarcity of pretrained subquadratic models with robust reasoning abilities, we develop recipes to distill specific reasoning skills into these architectures. We then benchmark the models for multiple Chain-of-Thought (CoT) completions, providing a comprehensive analysis of performance under fixed compute and memory constraints. Our approach advances the Pareto front established by existing models, achieving a better trade-off between efficiency and reasoning capability.

Our distilled pure and hybrid subquadratic reasoners are able to outperform their Transformer teachers on both coverage and accuracy on MATH (Lightman et al., 2023) and GSM8K (Cobbe et al., 2021) mathematical reasoning tasks on most time budgets, reaching the same quality with 2.5× less inference time. Our results highlight the potential for distilling reasoning and math capabilities across architectures in a cost-effective manner while maintaining the inference compute scaling properties of Transformers.

## 2 Related Work

### 2.1 Scaling Inference Time Compute for Reasoning

Scaling inference time compute has emerged as a promising strategy to improve the performance of LLMs. Techniques such as Chain of Thought (CoT) and its variants have demonstrated significant performance improvements across various reasoning benchmarks by decomposing complex tasks into intermediate steps (Wei et al., 2023; Yao et al., 2023)

While these approaches improve reasoning through task decomposition, they also increase computational demands due to longer generation sequences. Recent work suggests that this additional compute may itself contribute to improved model abilities (Pfau et al., 2024). Dynamic compute allocation during inference has further advanced this paradigm. For instance, Goyal et al. (2024) introduced pause tokens into the LLM vocabulary, enabling models to allocate compute effectively and achieve better reasoning and task performance.

Another prominent approach involves generating and searching multiple model outputs to select the best response. Various sampling algorithms have been proposed to increase the diversity and quality of generations to increase the likelihood of the best answer being selected (Wang et al., 2023; Renze & Guven, 2024; Zhang

et al., 2023). In parallel, outcome and process reward models (ORMs and PRMs) have been introduced to evaluate the quality of generated outputs and help guide intermediate generation steps within the LLM (Lightman et al., 2023; Zhang et al., 2024a; Luo et al., 2024; Uesato et al., 2022).

Recent work has shown that smaller LLMs, when scaled through inference-time compute (e.g., via majority voting or PRM-guided search), can outperform larger models under fixed compute budgets (Snell et al., 2024; Wu et al., 2024; Beeching et al., 2024). However, these findings are primarily limited to Transformer-based architectures. The extent to which these scaling laws apply to subquadratic architectures, which offer faster inference but may trade off expressiveness, remains underexplored.

## 2.2 Subquadratic Architecture Alternatives

While Transformers dominate the landscape of reasoning models (Grattafiori et al., 2024; Qwen et al., 2025), alternative architectures have been proposed to mitigate their high computational cost. These models, based on RNNs (Beck et al., 2024; Peng et al., 2023), SSMs (Gu et al., 2022; Gu & Dao, 2024), and linear attention mechanisms (Katharopoulos et al., 2020; Yang et al., 2024), offer improved inference and memory efficiency especially for long-context tasks and large-batch generation, making them attractive for large-scale language modeling. Notably, the Mamba family of models (Mamba-1 and Mamba-2) introduced selective state spaces, enabling linear-time sequence modeling without sacrificing performance (Gu & Dao, 2024; Dao & Gu, 2024). Hybrid architectures that combine subquadratic layers (e.g., Mamba) with a limited number of self-attention layers have since emerged, achieving superior performance compared to pure Transformer or subquadratic models (Lieber et al., 2024; Ren et al., 2024; Dong et al., 2024). Our work evaluates the inference-time scaling properties of both pure and hybrid subquadratic models, which are particularly well-suited for the increased compute demands of inference-time scaling

## 2.3 Knowledge Distillation

Knowledge distillation has proven effective in transferring capabilities from large teacher models to smaller, more efficient student models (Hinton et al., 2015). In the context of LLMs, distillation is commonly used to compress a larger pre-trained LLM into a smaller version while maintaining core knowledge and functionality (Gu et al., 2024; Xu et al., 2024). Although larger models exhibit better reasoning and overall abilities due to the properties of scale (Xu et al., 2025; Wei et al., 2022), distillation has enabled smaller models to achieve strong reasoning performance (DeepSeek-AI et al., 2025; Labs, 2025). While most distillation efforts focus on within-architecture transfer (e.g., Transformer to Transformer), recent work has explored cross-architecture distillation. Pretrained Transformers have been successfully distilled into recurrent architectures such as RNNs (Kasai et al., 2021; Mercat et al., 2024), linear attention (Zhang et al., 2024b), convolutions (Ralambomihanta et al., 2024), and SSMs (Bick et al., 2024; Wang et al., 2025). Whether strong reasoning can be distilled across architectures remains an open question.

## 3 Distilling Student Reasoners

In this section, we describe how we distill Llama models into pure Mamba and hybrid architectures. We refer to our pure Mamba models as **Llamba**, and our hybrid models as **MambaInLlama**. We distill both our hybrid and pure Mamba models using Llama 3.2-1B-Instruct and Llama 3.2-3B-Instruct from the Llama family of models (Grattafiori et al., 2024). Full details of the distillation algorithms can be found in Appendix A.

### 3.1 Distilling into Llamba

Our pure Mamba-distilled models, Llamba-1B and Llamba-4B, dubbed after Bick et al. (2025), which uses a similar methodology and the same teacher models, are distilled from their respective teacher models using our adjusted MOHAWK distillation approach (§ A) with only 8 billion tokens total each. Following Bick et al. (2024), we use a variant of Mamba-2 that converts the SSM head structure to multi-head (compared to the original multi-value and Llama's grouped-query structure) and converts the sequence mixer to entirely

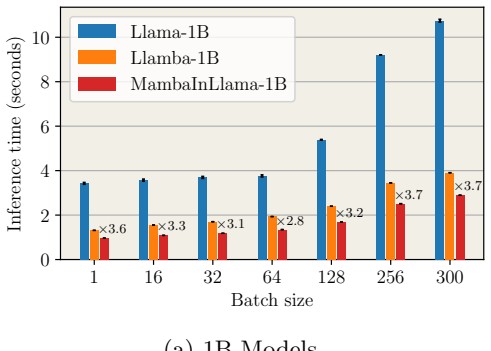
(a) 1B Models.

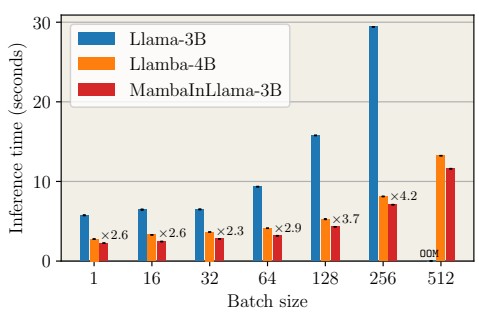
(b) 3B Models.

Figure 2: **Faster generation of distilled models.** In **(a)** and **(b)**, we show the inference time measured for the baseline Llama, our distilled Llamba (pure Mamba), and MambaInLlama (hybrid) models at the 1B and 3B scale. We denote the speedup for our MambaInLlama model at each batch size. We use prompts of 512 tokens and measure the time, not including the prefilling of the prompt, required to generate 512 tokens. Overall, distilled models can generate tokens much faster with the speedup being greater for larger batch sizes. Moreover, our distilled models are more memory efficient, as shown in **(b)**, using a batch size of 512 yields an Out of Memory (OOM) error for Llama 3B, but not for our models. To obtain 512 completions with Llama-3B, the two batches of 256, result in an inference time of 58.8s. In comparison, our MambaInLlama model would take 11.6s, a speedup of $\times 5.1$.

discrete-time. The post-convolution activation and pre-output projection normalization are also removed. The 8B token distillation dataset is composed of 4B tokens from FineMath-4+ (Lozhkov et al., 2024), allocated as 1B and 3B to Stages 1 and 2 respectively, and 4B tokens from OpenMathInstruct-2 (Toshniwal et al., 2024) used in Stage 3, which is the only stage in which we apply the chat template to the inputs. Unlike for our hybrid models, we find that computing the loss on both the assistant output and user prompt improves model performance over just the assistant output. All three distillation stages use the AdamW optimizer with $\beta = (0.9, 0.95)$ and weight decay of 0.1 and a Warmup-Stable-Decay (WSD) scheduler with 10% warmup and 10% decay (Hu et al., 2024) at a 2048 context length. In Stages 1 and 2, we set the learning rate to $1 \times 10^{-4}$, while in Stage 3, it is set to $1 \times 10^{-5}$. The hyperparameters are the same for the 1B and 4B distillation runs.

## 3.2 Distilling into MambaInLlama

Our hybrid Mamba models, named MambaInLlama-1B (with 4 attention layers in 16 total layers) and MambaInLlama-3B (with 6 attention layers in 26 total layers), are distilled with 8B tokens from OpenMathInstruct-2 (Toshniwal et al., 2024). We apply the Llama chat template, mask the user prompt, and compute the loss only over the tokens generated in the assistant's output. Thus, the total number of supervised tokens is reduced to roughly 7B. To speed up training, we use data packing to merge different sequences into a single one until we reach the maximum sequence length which is set to 8192. We use the AdamW optimizer with learning rate $2 \times 10^{-5}$, $\beta = (0.9, 0.95)$ and a weight decay of 0.1. The hyperparameters are the same for the 1B and 3B models. In Mamba layers, we set the SSM state size to 16. Consequently, the number of SSM groups after expansion is $2048/16 = 128$ for the 1B model and $3072/16 = 192$ for the 3B model.

**Remarks on the distillation dataset.** We finetune the Llama teacher models on the same data used during distillation to avoid our models potentially gaining an unfair advantage. The results, reported in Figure 3, show that the continuous training of the base model on the distillation data mix has a negligible effect on performance. Moreover, we find that the selection of data used during distillation has a significant impact on the final capabilities of the distilled models. Switching the Stage 3 dataset in Llamba-1B from OpenMathInstruct-2 to OpenHermes-2.5 (Teknium, 2023) decreased greedy decoding accuracy on MATH (acc@1) by more than 10 percentage points.

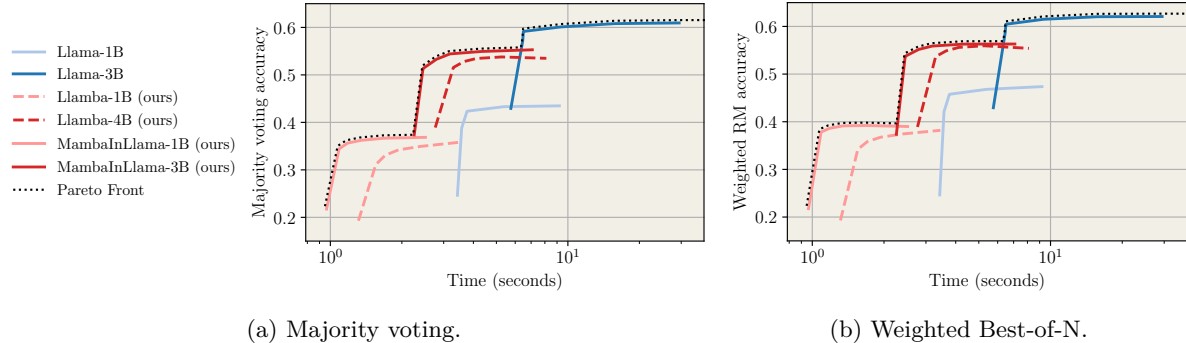

(a) Majority voting.

(b) Weighted Best-of-N.

Figure 4: **Distilled models provide better accuracies on MATH for most time budgets.** Figures are similar to Figure 1b. In **(a)** and **(b)**, we show the majority-voting accuracy and the weight Best-of-N accuracy (the selected answer is the one with the highest sum of reward model scores, as introduced in Beeching et al. (2024)), for different time budgets. Similarly to Figure 1b, we observe how the higher throughput of our distilled models allows them to push the Pareto front for most time budgets. Interestingly, when comparing models of a given size, Llama models are better for larger time budgets. However, looking at both model sizes together reveals that larger distilled models can compensate for the lower accuracies of smaller models. As a result, the Pareto front is defined by our hybrid models. While Llama-3B is still more efficient for a large time budget, one could imagine distilling a larger subquadratic model that generates quicker nonetheless.

**Lack of correlation between reasoning and general benchmarks.** We also highlight the lack of correlation between common multiple choice-based benchmarks and mathematical reasoning, as the OpenHermes variant of Llamba-1B outperforms the final Llamba-1B by more than 5 points on MMLU (Hendrycks et al., 2021a) and 0.5 point on ARC-Challenge (Clark et al., 2018). Moreover, analyzing the acc@1 performance of Llamba-1B and Llamba-4B on MATH after each of the three stages, we see that the sharp increase in reasoning ability between Stage 2 and Stage 3 is not reflected in general knowledge benchmarks (Fig. 12).

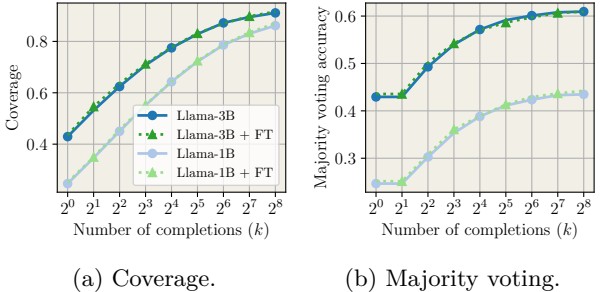

(a) Coverage.

(b) Majority voting.

Figure 3: **Negligible effect of finetuning Llama baselines on distillation dataset.** As our distillation dataset includes math content, we also finetune Llama models on the distillation dataset OpenMathInstruct-2. Those models are marked with the "+FT" prefix. We plot the coverage **(a)** and majority voting accuracies **(b)** as a function of the number of completions. We observe that finetuning Llama models on the distillation dataset has a negligible effect on those metrics.

### 3.3 Improving performance after distillation

We show that it is possible to improve the accuracy and coverage of our distilled models by performing some supervised fine-tuning (SFT) after distillation. This is inspired by Wang et al. (2025), where SFT is an integral part of the distillation protocol. Starting from the distilled MambainLlama-1B and 3B, we fine-tune the models for two epochs using 8 billion tokens from OpenMathInstruct-2. The distilled models achieve impressive performance both in coverage and accuracy, even surpassing the original Llama models. This is illustrated in Figure 7.

## 4 Scaling Inference Time Compute

We scale test-time compute using our distilled models by generating multiple CoTs to solve a set of math problems. The system prompt (Figure 11) contains instructions on how to properly format the response. The model outputs are parsed to extract the final solution, which is then compared to the ground truth. This approach enables us to evaluate the model's performance in generating correct solutions across multiple

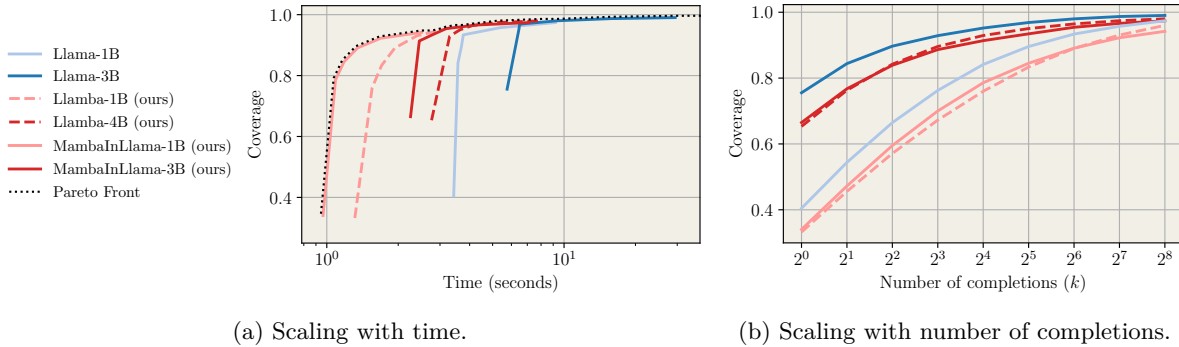

(a) Scaling with time.

(b) Scaling with number of completions.

Figure 5: **Distilled models have better coverage on GSM8K for most time budgets.** Observations are similar to Figure 1, despite a small degradation in coverage per number of completions, the better throughput of distilled models pushes the Pareto front forward. Hybrid models are more efficient for most time budgets.

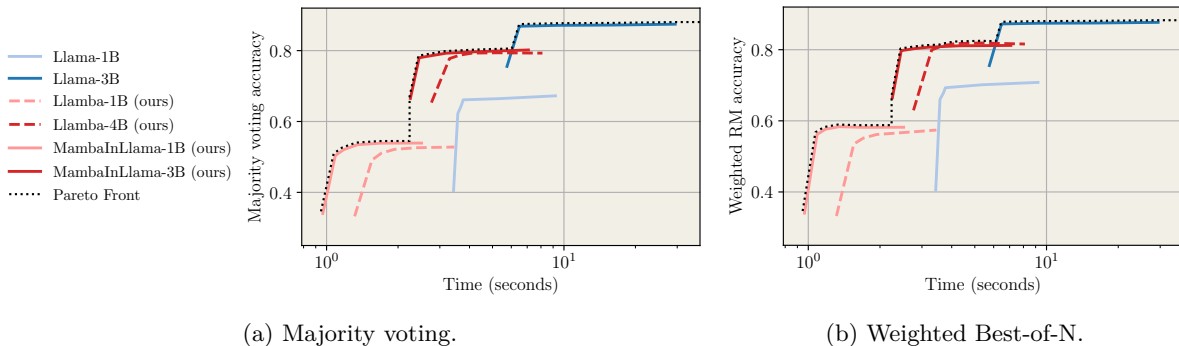

(a) Majority voting.

(b) Weighted Best-of-N.

Figure 6: **Distilled models provide better accuracies on GSM8K for most time budgets.** Observations are similar to Figure 4. The higher throughput of our distilled models allows them to push the Pareto front for most time budgets.

attempts. Moreover, the results demonstrate that the models are able to retain their instruction following ability after distillation.

**Evaluation metrics.** We evaluate our model using two primary metrics: coverage and accuracy. In domains like coding and formal proofs, where answers can be automatically verified, coverage directly translates to improved performance and has been widely adopted (Chen et al., 2021; Brown et al., 2024). Coverage is commonly referred to as the pass@k metric, where $k$ denotes the number of samples per problem (Chen et al., 2021; Brown et al., 2024). This metric estimates the probability that at least one correct solution exists among the $k$ samples. To reduce the variance when calculating coverage, we adopt the unbiased estimation formula from Chen et al. (2021). Specifically, we generate $N \geq k$ total samples per task. The probability that a correct solution exists among a pool of $k$ generated samples can then be determined given the total number of correct solutions $C_i$ for each task.

$$\text{pass@k} = \frac{1}{\# \text{ of problems}} \sum_{i=1}^{\# \text{ of problems}} \left( 1 - \frac{\binom{N-C_i}{k}}{\binom{N}{k}} \right)$$

We implement this formula using a numerically stable approach as suggested by Chen et al. (2021)(see Appendix D).

For accuracy, we use multiple aggregation strategies. Majority voting, or self-consistency decoding (Wang et al., 2023), is the most straightforward method of aggregating responses and computing an accuracy score. A more refined strategy involves using a trained verifier to select the best response (we call this approach Best-of-N).

For our verifier, we utilize a reward model trained using process supervision, where the model receives feedback on each step of the reasoning process. Inspired by Snell et al. (2024), we utilize a Llama-3.1 8B-based reward model to score the solutions for Best-of-N. As PRMs produce a cumulative sequence of step-level scores per solution, we perform a reduction over the steps to obtain a single solution-level score which we use for answer selection. Following Snell et al. (2024), we use the final score in all the steps as the score for Best-of-N.

While Best-of-N simply selects the generation with the highest reward, weighted Best-of-N aggregates the rewards of samples that share the same response and returns the solution with the highest combined score. In this work, we chose to report weighted Best-of-N, since we often found it to be superior to Best-of-N, most likely due to it identifying highly-rated but also common solutions. Inspired by the scaling laws for inference-time compute (Brown et al., 2024), we speculate that having a model that can scale to very large number of completions can be very effective, even more so than increasing model size by a large amount (Snell et al., 2024), so focus on the performance across fixed compute budget.

## 5 Results

We (i) measure the inference speedup of our distilled models (§ 5.1), and (ii) show that this speedup can result in better scaling for a given inference time budget (§ 5.2).

### 5.1 Inference time results

**Experimental protocol.** To measure the inference time and throughput of our distilled models, we focus on creating a realistic setup that matches the prompt and CoTs lengths that we find for MATH and GSM8K. We compare the runtime of a Llama 3.2 architecture against our distilled models. For Llama, we use FlashAttention2 (Dao, 2023) and torch compile. The implementations of our models rely on the standard Mamba implementation. Given a varying batch size, we consider the tasks of generating 512 tokens from a 512 token prompt, which reflect the length found in the evaluation datasets. The prefilling time to process the prompt is not included in the benchmark, as it may depend on the redundancy of a given prompt within the batch. In fact, in our setting, we are only interested in the time to generate the multiple completions given one prompt. Our benchmark is done on a single NVIDIA H100 GPU, and averaged results over multiple runs are shown in Figure 2.

**Distilled models are significantly faster.** Results in Figure 2 show our distilled models are up to $\times 3.7$ and $\times 4.2$ faster than their respective Llama 1B and 3B baselines. Moreover, MambaInLlama and Llamba are more memory efficient and thus can run larger batches. In Figure 2b, our models can accommodate batches of 512 while Llama-3B returns an out-of-memory error. We also notice that MambaInLlama models are slightly faster than Llamba. We speculate that this is because MambaInLlama has a smaller SSM state size of 16, while Llamba uses a larger SSM state size of 64. Additionally, Llamba has larger projections because of its multi-head structure.

**Remark.** In our speed experiments, both the prompt and the generation lengths are relatively short compared to many other domains. When evaluating multi-turn conversations, for example, where distilled Mamba architectures are already shown to excel (Wang et al., 2025), it is clear that the length of the context and of the CoTs can be significantly larger than what has been considered in our experiments. Such longer sequences would significantly increase the throughput advantage of our models. Unfortunately, it is not yet clear how to evaluate and exploit test-time compute for more subjective, conversational tasks.

**Limitations.** We try to ensure fair speed comparisons between the model types by utilizing roughly equivalent implementations: standard implementation of Mamba-1/2 and FlashAttention-based self-attention with Pytorch compilation. However, it is worth noting that there exist optimizations for current Transformer architectures, such as memory management systems like vLLM (Kwon et al., 2023), that enable faster

generation, yet similar improvements are currently not readily available for alternative architectures. Deeper optimizations for each architecture are beyond the scope of this work.

## 5.2 Results on reasoning tasks

**Experimental protocol.** We benchmark the teacher models (Llama-3.2 1B-Instruct and 3B-Instruct) and the distilled Mamba students (MambaInLlama-1B,-3B and Llamba-1B,-4B) on the MATH-500 subset of the MATH dataset (Lightman et al., 2023; Hendrycks et al., 2021b) and a randomly selected 500-sample subset of GSM8K (Cobbe et al., 2021). We evaluate performance based on coverage and accuracy with majority voting and weighted Best-of-N (§ 4). We sample responses from the distilled models with temperatures $T = 0.6, 0.8$ and top_k $= -1$ to consider all tokens. For each response, we sample up to 2048 tokens. For the distilled models, we find that $T = 0.6$ and $T = 0.8$ are ideal for the 1B and 3B scale, respectively, and subsequent results are obtained using these temperatures. The official Llama evaluation system prompt is used for the MATH dataset, while the original prompt is kept for GSM8K as displayed in Figure 11. For our process reward model, we utilize a base Llama3.1-8B-Instruct model trained on Mistral-generated data (Xiong et al., 2024).

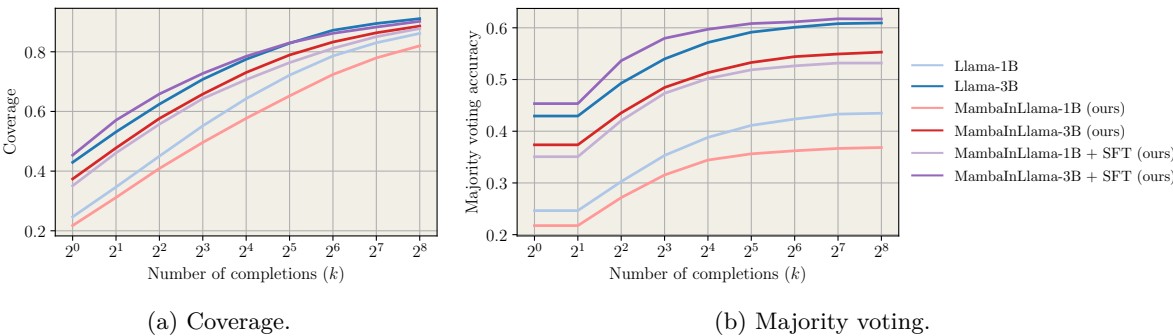

(a) Coverage.                     (b) Majority voting.

Figure 7: **SFT can further improve distilled models.** In **(a)**, and **(b)**, we show the coverage and majority voting accuracies for MambaInLlama distilled models. We show how coverage and majority voting accuracies can be further improved through Supervised Fine-Tuning (SFT). The improvement is especially striking for our 1B model.

**Distilled Models can Cover like Teachers.** When observing the scaling of coverage as the number of generation $k$ increases in Figure 1a (resp. Fig. 5a for GSM8K), we see distilled models closely matching the coverage of their teachers. Only a small degradation is observed. When we plot the coverage as a function of the time budget, by associating each coverage measurement to the time required to generate the associated number of completions (see Fig. 1b), we find that our distilled models are exceptional in their ability to generate correct answers fast. By generating many more completions for the same time budget, the overall Pareto front for coverage in both MATH and GSM8K (Fig. 1b, 5b) is heavily dominated by our distilled models where both pure and hybrid Mamba reasoners are able to achieve the same degree of coverage in nearly half the time of their respective teachers. Given a sufficiently strong reward model or verifiable solutions, those coverage scores would in large parts translate to accuracy.

**Distilled Models Achieve Competitive Accuracy Under Fixed Time.** Training with distillation and utilizing subquadratic architectures—which are less expressive than Transformers—can degrade model quality. However, we find that this is a worthy trade-off when comparing performance under fixed time budgets in Figure 6 and Figure 4.

Similarly to coverage, the lighter and faster batch inference of the distilled models, allowing for more generations, results in a better accuracy/time Pareto front at several completion scales. Interestingly, while comparing models of similar sizes indicates that larger time budgets are dominated by the teacher models, we observe that the larger distilled model can provide better accuracy than the smaller baseline while still being faster. For example, while Llama-1B provides better accuracy than MambaInLlama-1B for larger time budgets, MambaInLlama-3B takes over where MambaInLlama-1B left off, providing a better accuracy than

Llama-1B for inference time. Similarly, we conjecture that it would be possible to distill a larger baseline model that would outperform Llama-3B, providing better accuracy for a given time budget.

**Larger Students Faster and Better than Smaller Teachers.** The core driving force behind the growing interest in subquadratic models is their computational efficiency. Such properties enable our distilled models of larger sizes (3B scale) to generate samples faster than even a smaller Transformer (1B). Our MambaInLlama-3B and Llamba-4B models outperform the slower Llama-1B baseline on coverage and accuracy while being faster. These inference speedups, rooted in the underlying architecture, allow larger and more capable models to be used in time-constrained environments, despite their increased number of parameters.

**Smaller models have great coverage.** When focusing on coverage, we observe in Figure 1b and Figure 5b that most of the Pareto front is occupied by 1B models. Contrasting those results with the majority voting accuracy results in Figure 4a and Figure 6a where the gap between 1B and 3B models is much more significant. An interpretation is that, while smaller models have the ability to generate the correct answer, the probability of generating it within a constrained number of samples increases with the model size. This finding has implications in choosing model size for tasks involving formal language where the answer is easily verifiable, such as coding and mathematical proofs. In those applications, coverage matters most, and smaller models may be preferred given their better time/coverage efficiency compared to larger models.

**SFT improves the models significantly** Following distillation, which establishes a strong foundation for our models, we observe that additional supervised fine-tuning (SFT) significantly enhances their performance, as illustrated in Figure 7. This suggests that while distillation effectively transfers knowledge from the teacher model, SFT further refines and aligns the model's capabilities, enabling it to become highly competitive. Our results indicate that by leveraging both distillation and SFT, subquadratic architectures can match or even surpass their Transformer teachers in absolute performance on reasoning tasks.

## 6 Conclusion

In our work, we investigate whether lower-complexity models can leverage their superior generation throughput to outperform similarly sized Transformers under a fixed computational budget. We focus on reasoning tasks where we can scale test-time compute to improve performance. Through extensive experimentation, we distill both pure and hybrid Mamba models at the 1B and 3B scales and evaluate their reasoning capabilities on mathematical reasoning benchmarks, where the ability of subquadratic models to quickly generate many completions enables them to take advantage of their scaling properties when increasing inference compute. When fixing memory and/or compute, our models achieve better coverage and accuracy for most time budgets compared to their Transformer, teacher counterparts. These findings highlight the potential of Mamba and other attention alternatives as strong substitutes to Transformers for tasks that benefit from scalable inference compute.

We hope this work inspires future work in pretraining subquadratic reasoners and further exploring their inference scaling properties. More research is required to determine the best way to distill reasoning capabilities across architectures, as performance remains highly sensitive to both data and distillation techniques. Moreover, since our distilled models demonstrate exceptional coverage, developing better reward models to better identify correct answers can close the accuracy gap. Finally, further investigation into scaling inference compute for conversational and subjective tasks would open to a scenario where lighter subquadratic models can achieve larger gains in performance and speed

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

## A   Distillation Techniques

We utilize two separate techniques for our pure and hybrid Mamba model distillation.

**Llamba Distillation method.** In order to distill pure Mamba models, we modify the MOHAWK distillation procedure introduced by Bick et al. (2024). MOHAWK is composed of three stages: 1) matrix orientation, 2) hidden state alignment, and 3) weight transfer and knowledge distillation. Stage 1 (matrix orientation) aligns the Mamba-2 model's SSM matrix mixer (Dao & Gu, 2024) with the teacher's self-attention matrix by minimizing the distance between the two matrices. Stage 2 (hidden state alignment) matches the student and teacher's layers' hidden state outputs. Both of these stages are run independently across layers to prevent previous optimization gaps from propagation through the model. This is done by setting the input of the student layer to be that of the previous teacher layer's output. Stage 3 (weight transfer and knowledge distillation) transfers the remaining, unoptimized parameters, e.g., MLPs, embeddings, and norms, and finetunes the complete end-to-end student model using a distillation loss on the student and teacher logits (Hinton et al., 2015). We deviate from the original MOHAWK paper by transferring and training the MLP weights and norms of each teacher decoder layer to the student during Stage 2 instead of Stage 3. This stems from the architectural differences between Phi (Li et al., 2023) (MLP and self-attention in parallel) and Llama (Grattafiori et al., 2024) (sequential self-attention and MLP). Stage 3 remains the same with fewer weights transferred. We note that our pure Mamba-2 models are slightly larger than their Transformer counterparts due to the pattern conversion from grouped-query to multi-head and the additional parameters found within the Mamba-2 layer, like the gating parameters.

**Mamba-in-Llama Distillation method.** For the hybrid models, we modify the protocol proposed by Wang et al. (2025) in order to distill some specific capabilities. These techniques have been shown to be effective for hybrid architectures. The Mamba-in-Llama framework (Wang et al., 2025) introduces a method for distilling hybrid Transformer-Mamba models by reusing weights from the attention layers. The linear projections for $\mathbf{Q}$, $\mathbf{K}$, $\mathbf{V}$ and $\mathbf{O}$ are initialized using the corresponding linear projections for $\mathbf{C}$, $\mathbf{B}$, $\mathbf{X}$ and $\mathbf{O}$ respectively. The only additional learned parameters in the new layers are the sampling rate $\Delta$ and the dynamic $\mathbf{A}$. These new parameters will control the constructed Mamba through the discretization function. Specifically, we take $\Delta \in \mathbb{R}^{N'}$ to discretize $\mathbf{B}_t, \mathbf{C}_t \in \mathbb{R}^{N \times 1}$ and obtain $\overline{\mathbf{B}}_t, \overline{\mathbf{C}}_t \in \mathbb{R}^{N' \times N \times 1}$. We directly reuse the MLP layers. Differently from Wang et al. (2025), we replace the attention layers with Mamba layers in a single round and finetune the whole model. For distillation, we employ token-level KL divergence. The full probability distribution of the student model, $p(\cdot; \theta)$, is trained to align with the full distribution of the teacher model, $p(\cdot; \theta_T)$, by minimizing the KL divergence across all possible next tokens at position $t$. Differing from (Wang et al., 2025), we use the reverse KL divergence, $D_{\mathrm{KL}}(p(\cdot; \theta) \parallel p(\cdot; \theta_T))$ instead of the forward KL divergence as the loss function, since reverse KL behaves more like mode-seeking and better mimics the

---

**Algorithm 1** Initializing MambaInLlama from Llama

---

1: **Shapes:** $B$ - Batch, $L$ - Length, $D$ - embed size,
        $N = D/\text{Heads}$, $N'$ - expand
2: **Input:** $\boldsymbol{o}_t$: (B, D)
3: **Output:** output: (B, D)
4: **New Params:** MLP, $\mathbf{A}$
5: **for** each head $\mathbf{W}^k, \mathbf{W}^q, \mathbf{W}^v, \mathbf{W}^o : (N, D)$
        expanding grouped KVs **do**
6:     **Head Parameter:** $\mathbf{A} : (N, N')$
7:     for all positions $t$:
8:       $\boldsymbol{x}_t : (B, N) \leftarrow \mathbf{W}^V \boldsymbol{o}_t$
9:       $\mathbf{B}_t : (B, N) \leftarrow \mathbf{W}^K \boldsymbol{o}_t$
10:      $\mathbf{C}_t : (B, N) \leftarrow \mathbf{W}^Q \boldsymbol{o}_t$
11:      $\Delta_t : (B, N') \leftarrow \text{MLP}(\boldsymbol{x}_t)$
12:      $\overline{\mathbf{A}}_{1:T}, \overline{\mathbf{B}}_{1:T}, \overline{\mathbf{C}}_{1:T} : (B, N, N') \leftarrow \text{DISC}(\mathbf{A}, \mathbf{B}, \mathbf{C}, \Delta)$
13:      $\boldsymbol{y} \leftarrow \text{LINEARRNN}(\overline{\mathbf{A}}, \overline{\mathbf{B}}, \overline{\mathbf{C}}, \boldsymbol{x})$
14:      output $\leftarrow$ output $+ \mathbf{W}^{O\top} \boldsymbol{y}$
15: **end for**
16: **return** output

---

peak values. We also find that it yields better results empirically. We adopt the Mamba-1 architecture for our hybrid models, as Lieber et al. (2024); Wang et al. (2024); Dong et al. (2024) demonstrates that using Mamba-1 in a hybrid architecture yields better results, especially for challenging reasoning tasks.

## B Additional results

Figures 8 and 9 show accuracy as a function of the number of completions.

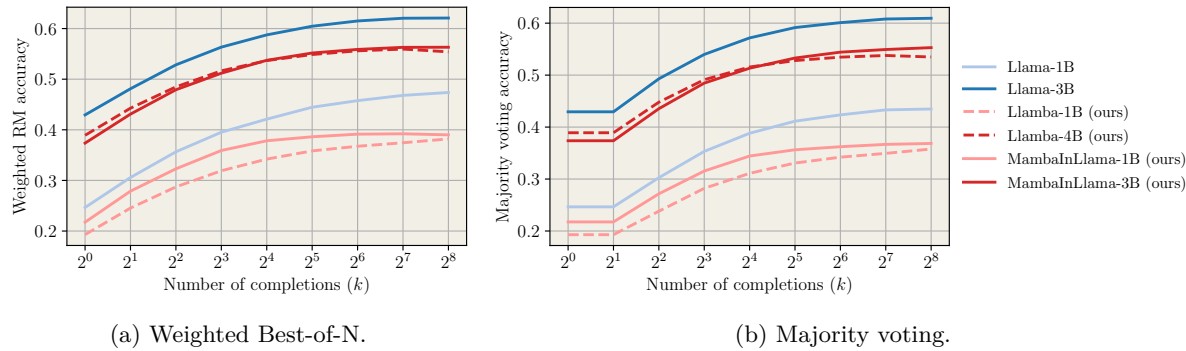

(a) Weighted Best-of-N.

(b) Majority voting.

Figure 8: **Accuracy scores on MATH per completion without considering inference time.**

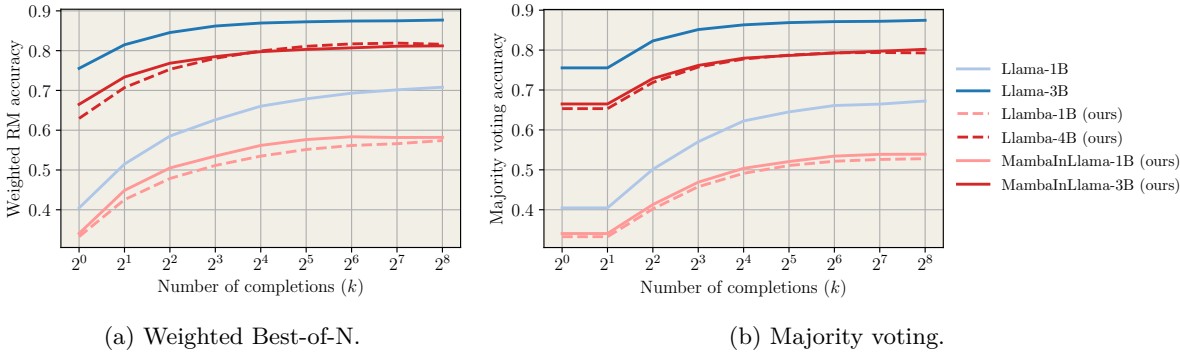

(a) Weighted Best-of-N.

(b) Majority voting.

Figure 9: **Accuracy scores on GSM8K per completion without considering inference time.**

## C Distillation loss curves

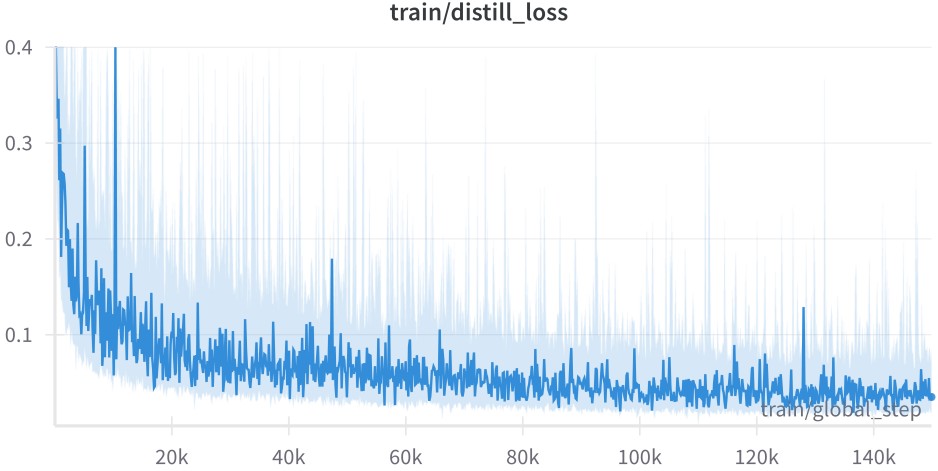

Figure 10: **The distillation loss decreases steadily over time, showing a clear downward trend and converge at around 0.03.**

Figure 10 shows the distillation loss of MambaInLlama 3B when training on the Toshniwal et al. (2024) dataset. We observed a stable decrease in loss until it converged to 0.03.

## D Pass@k implementation

A numerically stable script for calculating an unbiased estimate of pass@k, from Chen et al. (2021).

```python
def pass_at_k(n, c, k):
    """
    :param n: total number of samples
    :param c: number of correct samples
    :param k: k in pass@$k$
    """
    if n - c < k: return 1.0
    return 1.0 - np.prod(1.0 - k /
                         np.arange(n - c + 1, n + 1))
```

## E System Prompts

**MATH System Prompt:** *Solve the following math problem efficiently and clearly:\n\n- For simple problems (2 steps or fewer):\nProvide a concise solution with minimal explanation.\n\n- For complex problems (3 steps or more):\nUse this step-by-step format:\n\n## Step 1: [Concise description]\n[Brief explanation and calculations]\n\n## Step 2: [Concise description]\n[Brief explanation and calculations]\n\n...\n\nRegardless of the approach, always conclude with:\n\nTherefore, the final answer is: $\boxed{answer}$. I hope it is correct.\n\nWhere [answer] is just the final number or expression that solves the problem.*

**GSM8K System Prompt:** *\n\nGiven the following problem, reason and give a final answer to the problem.\nYour response should end with "The final answer is [answer]" where [answer] is the response to the problem.\nProblem:*

Figure 11: **System prompts for MATH and GSM8K.** System prompts passed to all models (pure and hybrid distilled models and Llama baseline) to be used within the chat template when generating solutions to questions in the MATH and GSM8K datasets.

# F MOHAWK (Pure Mamba) Distillation on Reasoning and General Performance

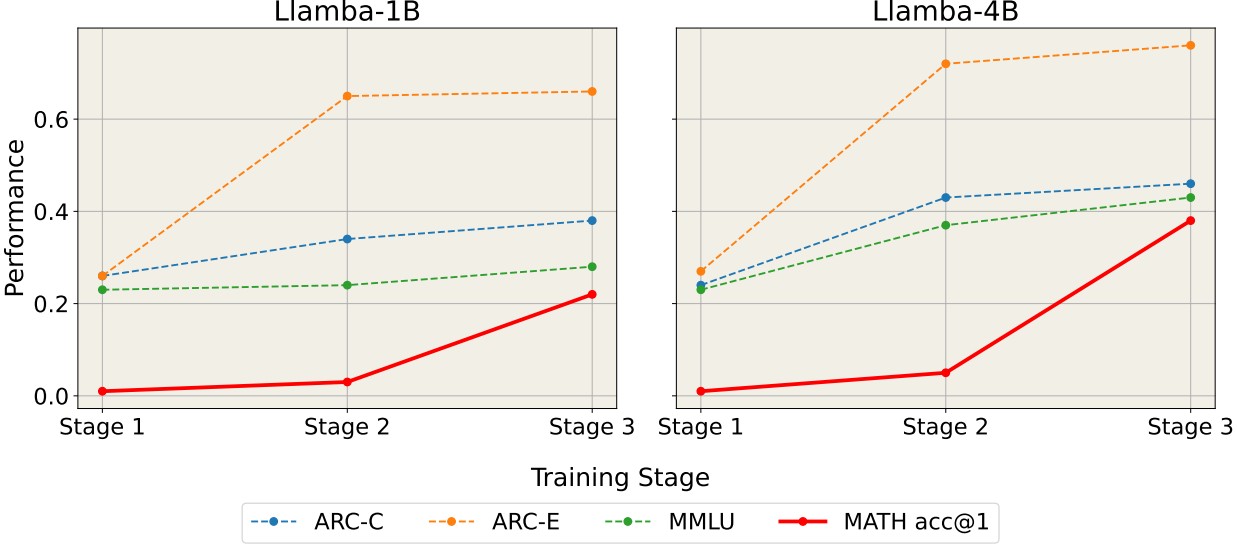

Figure 12: **Reasoning and general performance of models after various stages of MOHAWK distillation.** Distilling reasoning capabilities, measured by accuracy using greedy decoding (acc@1) on the MATH benchmark, does not correlate with general multiple-choice reasoning benchmarks. The performance of Llamba-1B and 4B increase significantly only after the last stage of the MOHAWK distillation, but Stage 2 is able to significantly improve general benchmark performance.

