# OpenReview forum: "Thinking Slow, Fast: Scaling Inference Compute with Distilled Reasoners"
_TMLR — Rejected by TMLR_

### Review · Reviewer_Uyvg · 2025-10-05

**Summary Of Contributions:**

This paper's novelty is in empirically proving that for reasoning tasks, a subquadratic model can outperform a more accurate Transformer by trading per-sample quality for a sheer quantity of attempts.

### Strengths
- The paper starts by identifying a clear, important, and practical motivation. The writing is easy to follow and clear.
- The experiments discussed in the paper are detailed and the transparency from the authors are much appreciated.
---

### Weaknesses
- Can the authors clarify why they use Llamba-4B (when the teacher models have lesser parameters)? The use of Llamba-4B raises fairness concerns, as any observed improvements may stem from its larger parameter count rather than genuine architectural or distillation advantages.
- The concepts of Llamba [1] and Mamba-in-Llama [2] have already been explored in prior work. Beyond the minor architectural modifications described here and while it's interesting to see a neat comparison between the above two and Llama, many of the conclusions in Sections 5.1 and 5.2 are either directly stated or can be inferred from experiments in these papers, which already demonstrates strong distillation performance and inference-time efficiency across batch sizes.
- While the Pareto frontier analysis and limited inference-time evaluations with coverage add value, the focus on math benchmarks is limiting. Given the paper’s stated motivation around improving reasoning capabilities, the authors should include broader reasoning benchmarks to substantiate claims beyond mathematical reasoning. This is a minor concern but a study on other benchmarks would be necessary.
- Considering Figures 8 and 9, the distilled models only outperform Llama under constrained inference-time settings. Under unlimited inference compute (which is also a practical setting to test reasoning capabilities), Llama achieves higher accuracy, suggesting that the proposed approach primarily trades accuracy for efficiency. The authors should therefore investigate methods that preserve both reasoning accuracy and CoT quality while maintaining efficiency.
- "Unfortunately, it is not yet clear how to evaluate and exploit test-time compute for more subjective, conversational tasks." While appreciated, could the authors clarify why they are unable to evaluate under these settings? These settings arguably represent the most realistic and practically relevant deployment scenarios.
---

[1] Bick et al. Llamba: Scaling Distilled Recurrent Models for Efficient Language Processing. arXiv:2502.14458.

[2] Wang et al. The Mamba in the Llama: Distilling and Accelerating Hybrid Models. arXiv:2408.15237v4.

**Audience:**

Yes

**Audience Explanation:**

The paper's exploration of distillation efficiency and inference-time tradeoffs would definitely interest readers focused on scalable reasoning and LLM optimization.

**Broader Impact Concerns:**

N/A.

**Claims And Evidence:**

No

**Claims Explanation:**

Most claims in the paper are supported by the comprehensive experiments given. However, several claims are insufficiently supported by novel evidence, as stated in the weaknesses above. There are a few other experiments I would recommend the authors to perform:
- An ablation study to empirically validate the contributions of the modified MOHAWK algorithm.
- Could the authors clarify the rationale for selecting the specific datasets for Stages 1, 2, and 3? Providing ablation studies on these choices would be appreciated (especially with reasoning datasets).

**Requested Changes:**

Several changes and questions have been given in the Weaknesses above. Regardless, here are some minor concerns I have:
- The paper's formatting could be improved, as figures are scattered throughout.
- What is Figure 3's benchmark? It isn't explicitly mentioned.

---

### Review · Reviewer_sXYt · 2025-10-12

**Summary Of Contributions:**

# Summary Of Contributions
This paper investigates whether lower-complexity models can leverage faster generation to outperform similarly sized Transformers under fixed computational budgets. To explore this, the authors distill both pure and hybrid Mamba models from pretrained Transformers, addressing the current lack of efficient subquadratic reasoners. The distilled models, trained on only 8 billion tokens, demonstrate scaling behavior and competitive performance on mathematical reasoning tasks, particularly excelling in inference speed for large batches and long sequences. While distillation introduces a drop in zero-shot performance, the Mamba variants surpass their Transformer teachers in coverage and accuracy under fixed time constraints.


# Strengths And Weaknesses

## Strengths
* The paper addresses a timely and relevant challenge in LLM research: how to optimize model architectures for efficient test-time compute during reasoning tasks, particularly under memory and compute constraints.

* The authors provide a practical approach and recipes to distill from pretrained Transformers into both pure and hybrid Mamba models.

## Weakness
* the paper lacks proper structuring and formatting of the methodology, evaluation, and analysis of results. There is lack of problem formulation, background and notation, methodological abstraction and mathematical derivations.

* the paper claim is broad but the experimental setup and results are narrow. Either the claim needs to be focused on the supportive evidence or the experimental results be expanded to cover the broad claim. Particularly, the experimental setup is on the MATH datasets, hence it is not supported how it would generalize to other reasoning problems.

* The evidence is only given on Llama 1B and 3B models while the claim is not limited to this range of model parameters. Again the evidence does not accurately support the claim. It is difficult to generalize from results on the particular language models of Llama.

* It seems obvious from the findings in the knowledge distillation literature that by distilling from higher reasoning capacity teacher models like quadratic Transformers into sub-quadratic models, the inference time would be improved while reasoning capabilities maintained.

* comparison of linear Transformers with the current results is missing. How do we know if linear approximation of the quadratic Transformers cannot achieve similar test-time compute efficiency results.

* There is limited discussion on the potential degradation in zero-shot generalization performance due to distillation, which could impact usability in open-ended or less structured tasks.

* Although sub-quadratic models are promoted for their efficiency, the paper does not extensively explore the trade-offs in training complexity, convergence behavior, or stability compared to Transformers.

* there are experimental details and guidelines on the distillation recipe all across the methodology chapter 3 and 4. They need to be limited to the experimental evaluation as they don't have methodological value to the paper.

**Audience:**

Yes

**Audience Explanation:**

Conditioned on the improvement suggested above, the materials presented in the paper are still practically insightful for some of the targeted audience.

**Broader Impact Concerns:**

There are no "Broader Impact Concerns" for this paper.

**Claims And Evidence:**

No

**Claims Explanation:**

It is very challenging to state the claims are supported as so at the current state of the work. Therefore, we need to revisit and re-evaluate this later on. At the current state, the paper needs extensive restructuring and reformatting of the content to help improve the narration of the claim/evidence for the audience. Additionally, there is a significant gap between the claim and the evidence as noted in the previous section.

**Requested Changes:**

Based on the aforementioned strengths and weaknesses, the requested changes are provided in an orderly manner from the highest (critical) to the lowest (nice to have) importance as follows:

* aligning the claim with the evidence and try to close the gap between the two ends by narrowing the claim or expanding the evidence.

* the paper needs to develop from the problem formulation and notation into the abstraction of the claim and the derivation of the methodolgy given the notation introduced earlier.

* restructuring the content of the paper such that the practicaly recipes and findings be limited and shortened into the experimental evaluation section.

* experimental results on; other families of models such as Qwen; other domains such as coding and general and commonsence reasoning problems, linear Transformer models, larger models (ie > 3 B)

---

### Review · Reviewer_7mTJ · 2026-02-21

**Summary Of Contributions:**

This work optimizes test-time compute for math reasoning by distilling Llama-3.2 into linear-complexity Mamba and hybrid models. Despite a minor drop in zero-shot accuracy, the distilled models exhibit high throughput, allowing them to outperform their Transformer teachers under fixed time budgets via majority voting. This result validates subquadratic architectures as offering a superior Pareto frontier for reasoning performance versus latency.

**Strengths**

* The paper effectively connects two high-impact research directions—inference-time compute scaling and subquadratic architectures—demonstrating that architectural efficiency can be traded for improved reasoning coverage.
* The distilled models achieve a superior Pareto frontier on MATH and GSM8K, outperforming teacher models in accuracy when constrained to the same inference time budget.
* The method achieves competitive performance using a highly efficient distillation protocol (only 8B tokens) followed by Supervised Fine-Tuning (SFT), making the approach accessible.
* The study correctly prioritizes performance under fixed compute/time budgets rather than raw accuracy, which is the relevant metric for real-world deployment of reasoning models.

**Weaknesses**

* It is unclear if the time budget calculations for the "Best-of-N" strategy include the inference time of the 8B Reward Model. If excluded, the practical speed advantage of the system is overstated.
* The comparison relies on standard attention mechanisms; it does not benchmark against Transformers utilizing Sparse Attention or Speculative Decoding, which could significantly narrow the speed gap without changing architectures.
* The paper does not investigate how the size of the Reward Model affects the total system latency, leaving open the question of whether a smaller, faster verifier could maintain the performance gains.
* The evaluation is restricted to mathematical reasoning (MATH, GSM8K), limiting the ability to generalize the findings to other domains such as coding or commonsense reasoning.

**Audience:**

Yes

**Audience Explanation:**

Researchers in the field of LLMs will be interested in fast inference.

**Claims And Evidence:**

Yes

**Claims Explanation:**

Most of the claims are supported by strong evidence, except for the time budget calculations and the comparison with efficient attention mechanisms.

**Requested Changes:**

1. Explicitly state in the text and captions for Figures 4(b) and 6(b) whether the inference time of the Reward Model is included in the total time budget. If it is excluded, this must be highlighted as a limitation, as it significantly affects the claim of superior system-level latency in a real-world deployment.
2. Expand the "Limitations" or "Related Work" section to qualitatively compare the proposed Mamba-based approach against Transformers optimized with Sparse Attention or Speculative Decoding. The authors should acknowledge how these techniques might narrow the latency gap compared to the standard FlashAttention baseline used.
3. Add a discussion regarding the impact of the Reward Model's size on the total inference budget. Specifically, address whether a smaller reward model (e.g., matching the student model's size) could be used to reduce verification latency without significantly degrading the selection quality.

---

### Decision · Action_Editor_xbvt · 2026-04-10

**Recommendation:** Reject

**Audience:**

Yes

**Audience Explanation:**

All reviewers agreed that the motivation of this work is clear and significant, as reflected in comments such as “The paper effectively connects two high-impact research directions,” “The paper addresses a timely and relevant challenge in LLM research,” and “The paper starts by identifying a clear, important, and practical motivation.” Nevertheless, the reviewers identified methodological limitations and weaknesses in the empirical evaluation.

**Claims And Evidence:**

No

**Claims Explanation:**

Reviewers identified several important issues that, if addressed, could substantially strengthen the paper.
These range from paper organization and presentation to the need for stronger empirical validation, including compute-efficiency analysis, comparisons with sparse attention and speculative decoding methods, and evaluation on additional reasoning benchmarks such as coding and commonsense reasoning tasks, to name a few. However, no author rebuttal or revised manuscript was submitted, leaving the reviewers' concerns unaddressed. Based on these assessments, the Action Editor concludes that the submission is not yet ready for publication.